# Modulation of miR-29a and miR-29b Expression and Their Target Genes Related to Inflammation and Renal Fibrosis by an Oral Nutritional Supplement with Probiotics in Malnourished Hemodialysis Patients

**DOI:** 10.3390/ijms25021132

**Published:** 2024-01-17

**Authors:** Corina Verónica Sasso, Said Lhamyani, Francisco Hevilla, Marina Padial, María Blanca, Guillermina Barril, Tamara Jiménez-Salcedo, Enrique Sanz Martínez, Ángel Nogueira, Ana María Lago-Sampedro, Gabriel Olveira

**Affiliations:** 1Servicio de Endocrinología y Nutrición, Hospital Regional Universitario de Málaga, 29009 Málaga, Spain; veronica.sasso@ibima.eu (C.V.S.); saidlhamyani@gmail.com (S.L.); franciscohs296@gmail.com (F.H.); mpadial27@gmail.com (M.P.); 2Instituto de Investigación Biomédica de Málaga IBIMA-Plataforma BIONAND, 29009 Málaga, Spain; 3Departamento de Medicina y Dermatología, Universidad de Málaga, 29010 Málaga, Spain; 4CIBER de la Fisiopatología de la Obesidad y Nutrición, Instituto de Salud Carlos III, 29010 Málaga, Spain; 5Servicio de Endocrinología y Nutrición, Hospital Universitario Rey Juan Carlos, 28933 Madrid, Spain; mblancamb@gmail.com (M.B.); enrique.sanzm@quironsalud.es (E.S.M.); 6Servicio de Nefrología, Hospital de la Princesa, 28006 Madrid, Spain; gbarril43@gmail.com (G.B.); noro1976@hotmail.com (Á.N.); 7Servicio de Nefrología, Hospital Regional Universitario, 29010 Málaga, Spain; tajimsal@hotmail.com; 8CIBER de Diabetes y Enfermedades Metabólicas Asociadas, Instituto de Salud Carlos III, 29010 Málaga, Spain

**Keywords:** chronic kidney disease, oral nutritional supplement, malnutrition, hemodialysis, probiotics, circulating miRNAs, gene expression, inflammation, renal fibrosis

## Abstract

Malnutrition is prevalent in patients with chronic kidney disease (CKD), especially those on hemodialysis. Recently, our group described that a new oral nutritional supplement (ONS), specifically designed for malnourished (or at risk) hemodialysis patients with a “similar to the Mediterranean diet” pattern, improved caloric-protein intake, nutritional status and biomarkers of inflammation and oxidation. Our aim in this study was to evaluate whether the new ONS, associated with probiotics or not, may produce changes in miRNA’s expression and its target genes in malnourished hemodialysis patients, compared to individualized diet recommendations. We performed a randomized, multicenter, parallel-group trial in malnourished hemodialysis patients with three groups (1: control (C) individualized diet (*n* = 11); 2: oral nutritional supplement (ONS) + placebo (ONS-PL) (*n* = 10); and 3: ONS + probiotics (ONS-PR) (*n* = 10)); the trial was open regarding the intake of ONS or individualized diet recommendations but double-blinded for the intake of probiotics. MiRNAs and gene expression levels were analyzed by RT-qPCR at baseline and after 3 and 6 months. We observed that the expression of miR-29a and miR-29b increased significantly in patients with ONS-PR at 3 months in comparison with baseline, stabilizing at the sixth month. Moreover, we observed differences between studied groups, where miR-29b expression levels were elevated in patients receiving ONS-PR compared to the control group in the third month. Regarding the gene expression levels, we observed a decrease in the ONS-PR group compared to the control group in the third month for *RUNX2* and *TNFα*. *TGFB1* expression was decreased in the ONS-PR group compared to baseline in the third month. *PTEN* gene expression was significantly elevated in the ONS-PR group at 3 months in comparison with baseline. *LEPTIN* expression was significantly increased in the ONS-PL group at the 3-month intervention compared to baseline. The new oral nutritional supplement associated with probiotics increases the expression levels of miR-29a and miR-29b after 3 months of intervention, modifying the expression of target genes with anti-inflammatory and anti-fibrotic actions. This study highlights the potential benefit of this oral nutritional supplement, especially associated with probiotics, in malnourished patients with chronic renal disease on hemodialysis.

## 1. Introduction 

Malnutrition is highly prevalent in patients with chronic kidney disease (CKD), particularly those on hemodialysis, and is associated with increased mortality rates, decreased physical function and poorer outcomes. The etiology of malnutrition is multifactorial and includes decreased protein-calorie intake, inflammation, hypercatabolism, physical inactivity, nutrient loss into the dialysate, metabolic acidosis, uremic toxicity and nutrient malabsorption, among others [1]. Recently, our group described that a new oral nutritional supplement (ONS), specifically designed for malnourished (or at risk) hemodialysis patients with a “similar to the Mediterranean diet” pattern (made out of functional nutrients such as extra virgin olive oil, n-3 fatty acids, whey protein, fiber and antioxidants), improved caloric-protein intake, nutritional status (especially fat-free mass) and some biomarkers of inflammation and oxidation. Furthermore, the addition of probiotics could act synergistically with the ONS components to improve these biomarkers [2].

MicroRNAs (miRNAs) are a class of small, single-stranded, non-coding RNA molecules that play an important role in the regulation of gene expression. Most miRNAs are transcribed from DNA sequences as primary miRNAs and processed into precursor miRNAs and finally mature miRNAs [3]. In general, miRNAs interact with the 3′ untranslated region (3′ UTR) of target mRNAs, causing mRNA degradation and translational repression, but under certain conditions, miRNAs can activate translation or regulate transcription [4]. miRNAs are very stable in human biological fluids, including blood and serum, as they are packaged in the blood vessel membrane (such as exosomes, microparticles and apoptotic bodies) and associated with RNA-binding proteins. Many studies highlight the role of circulating microRNAs as biomarkers for several diseases [5]. miRNAs are involved in normal kidney function and development. They are also implicated in several renal diseases, including CKD, where patients show specific circulating miRNA expression profiles associated with the regulation of metabolic, muscle and inflammation functions and specific miRNAs have been identified as key players in the fibrosis process [6,7]. The expression of miRNAs is potentially altered by external factors such as diet [8,9] or the microbiota [10]. These miRNA profiles do not seem to vary before or after dialysis because most of them are not removed by hemodialysis membranes [11,12]. This makes miRNAs potential biomarker candidates even in this stage of kidney disease. Therefore, it would be interesting to evaluate the effect of a nutritional intervention on protein energy malnutrition associated with CKD. For example, miR-21 is overexpressed in CKD in animal models and humans. It also regulates metabolic pathways involved in fibrogenesis, inflammation, oxidation–reduction activity and cell proliferation [13,14]. Recent studies have described the role of miR-29 family as a strong regulator in several diseases such as obesity and diabetes [15] and a regulator of lipids metabolism during renal fibrosis [12]. Specifically, miR-29a and miR-29b have been associated with various protective effects in renal disease, particularly in the context of fibrosis, inflammation, apoptosis and kidney injury [16,17]. Furthermore, circulating levels of mir-155 (related to inflammatory pathways) and miR-126 (involved in vascular homeostasis) are reduced in patients with hemodialysis [12,13,18]. Other miRNAs have been related with CKD, such as miR-128a, which intervenes in mechanisms of muscle atrophy [19], while miR-223 intervenes in inflammation processes and is associated with vascular complications [18]. Finally, miR-378 is involved in angiogenic processes and adipogenesis, and it can be modulated by inflammatory cytokines such as TNFalpha, IL6 and leptin [20]. 

MiRNAs have been suggested as potential diagnostic and therapeutic tools for CKD; however, further research is needed to fully understand the role of miRNAs in CKD and their potential as a biomarker or therapeutic target [21,22]. Furthermore, it is known that miRNA levels change in response to different dietary interventions [23]: miRNA levels can be regulated by carbohydrates, protein, fat, vitamins, minerals, dietary fiber or even isolated nutrients or bioactive compounds [9,24]. Further studies are needed to fully understand the mechanisms underlying these interactions and to develop effective strategies for using oral nutrition supplements to modulate miRNA expression. Our aim in this study was to evaluate whether the new ONS, associated with probiotics or not, may produce changes in miR-21, miR-29a, miR-29b, miR-126, miR-128, miR-155, miR-223 and miR-378 expression and their target genes in malnourished hemodialysis patients, compared to individualized diet recommendations.

## 2. Results

A total of 31 patients (11 corresponding to group C, 10 to ONS-PL and 10 to ONS-PR) completed the 6-month trial (Figure 1) [2]. 

At the baseline visit, there were no basal significant differences between groups regarding age, sex, diabetes, Charlson comorbidity index or intake of fermented milk or antibiotics during the month prior to inclusion (Table 1). Furthermore, there were no baseline differences in any of the parameters for the nutritional assessment, dietary intake or biochemical data [2]. The two groups that took supplements significantly increased their weight and fat-free mass (FFM) both at 3 and 6 months. We also observed a slight decrease in extracellular water measured by bioimpedance, but without significant differences between groups or to the baseline value. Moreover, all these body composition changes occurred without an increase in serum phosphorus or potassium levels (with a clear tendency to decrease in the supplemented groups) [2].

### 2.1. Circulating miRNAs’ Expression Levels before and after Intervention

The changes in miRNAs’ expression levels are represented in Figure 2. 

There were no differences at baseline in the expression of any of the miRNAs studied between groups. We found a significant increase in the miR29-a expression in the ONS-PR group at the 3-month intervention compared to baseline (*p* = 0.03). 

Regarding miR29-b expression levels, we found an increase in the ONS-PR group at the 3-month intervention compared to baseline (*p* = 0.023) and compared to the control group (*p* = 0.027). 

We did not find significant differences between the groups and between times of intervention for miR-21, miR-126, miR-128, miR-155, miR-233 and miR-378 expression levels. 

### 2.2. In Silico Identification of Predicted and Validated miR-29a and miR-29b Target Genes

The bioinformatic analysis has revealed 2900 potential TGs for miR-29a and 2600 miR-29b. Moreover, the Panther and DAVID databases have shown the implication of the examined miRNAs in various biological processes. We accorded priority to TGs and biological processes linked with our ONS and renal function including inflammation, lipid metabolism and bone formation (Table 2, Figure 3). Then, we chose nine important TGs that regulate two or more of selected processes (highlited in bold in Table 2). 

### 2.3. Gene Expression Levels from Human Blood before and after Intervention

The changes in miR-29a and miR-29b target genes’ expression are represented in Figure 4. 

*PTEN* expression was increased at the 3-month intervention compared to baseline in the ONS-PR group (*p* = 0.023) and had a slight but not significant increased expression compared to the control group (*p* = 0.080). We found no differences between *PTEN* expression at the 6-month intervention and baseline. 

The expression of *TGFB1* was decreased at the 3-month intervention compared to baseline in the ONS-PR (*p* = 0.019) and ONS-PL group (*p* = 0.027). We observed an increase in *TGFB1* expression at the 6-month intervention in the ONS-PL group, when compared to the 3-month intervention (*p* = 0.016). 

Regarding RUNX2 expression, we observed a decrease in the ONS-PR group compared to the control group at the 3-month intervention (*p* = 0.012). In the ONS-PR group, we observed a slight decrease in *RUNX2* expression at the 3-month intervention compared to baseline (*p* = 0.079) and a later increase at the 6-month intervention (*p* = 0.023). 

The *LEPTIN* expression was increased at the 3-month intervention compared to baseline in the ONS-PL group (*p* = 0.04). We found no significant differences between the 6-month intervention and baseline. 

Regarding *TNFa* expression, we observed a decrease in the ONS-PR group at the 3-month intervention compared to the control group (*p* = 0.041). Furthermore, we observed a slight but not significant decrease in the ONS-PR group at the 3-month intervention compared to baseline (*p* = 0.098).

We also measured *AKT1*, *RELA*, *IL1B*, *IL6*, *MYH7B*, *HEF2C*, *CTNNB1*, *GHRL*, *RASA1* and *PPARG* expression gene levels, for being target genes of the studied miRNAs, but we found no significant differences between the groups and the times of intervention.

### 2.4. Correlations between miRNA Levels with Serum Biomarkers and Expression Levels of Target Genes

We have previously described significant changes in some inflammation biomarkers in the groups on supplements during the intervention [2]. Now, we analyzed the correlations between the percentage change between these variables and miRNA levels, represented in Figure 5.

When comparing baseline and the 3-month intervention, we found a significant negative correlation between the expression of miR-29b with TNFalpha (*p* = 0.017) in the ONS-PL group. In the ONS-PR group, we found a significant positive correlation between miR-29a and IL-10 (*p* = 0.025) and a significant negative correlation between miR-29b with IL-8 (*p* = 0.020). 

Regarding the correlations between miRNA levels and expression levels of target genes, we did not find significant results. 

## 3. Discussion

In this study, we analyzed for the first time the effects of a new oral supplement combined or not with probiotics on miRNA expression and its target genes in malnourished hemodialysis patients. The results provide valuable insights into the intricate relationship between nutritional interventions, miRNA regulation and potential therapeutic outcomes. Notably, we observed that the expression of miR-29a and b increased significantly in patients with ONS-PR at 3 months in comparison with baseline. Moreover, miR-29b expression levels were elevated in patients receiving ONS-PR compared to control at 3 months. We also found that the expression of its target genes related to inflammation and renal fibrosis was modified, especially in the group that received probiotics. Our previous study had shown that the specific ONS improved intake and nutritional status (mainly fat-free mass) and was associated with a reduction in biomarkers of inflammation and oxidation; the addition of probiotics could have had an effect a synergistic effect with ONS [2]. It is well known that miR-29a and miR-29b are implicated in the pathophysiology of various diseases, including obesity, diabetes, cardiovascular diseases and cancers [15,25,26,27]. Also, recent studies highlighted the protective effects of miR-29a and miR-29b against chronic kidney disease (CKD) and renal fibrosis, which are characterized by excessive accumulation of collagen and fibronectin in the extracellular matrix (ECM), inflammation and oxidative stress [28,29,30,31]. Moreover, in the present study, we revealed changes in *TNFα* expression, a positive correlation between miR-29a and miR-29b with anti-inflammatory markers such as IL-10 and a negative correlation between miR-29b with pro-inflammatory markers TNFalpha and IL-8. All these findings may underscore the potential anti-inflammatory effects of the oral supplement and probiotics within miR-29a and miR-29b regulation. The decrease in *TNFα* expression aligns with the notion that these supplements might contribute to decreasing the chronic inflammatory state associated with malnutrition and renal dysfunction. Specially, the decreased *TNFα* expression observed in the ONS-PR group could be indicating a possible synergistic nutrition-probiotic effect. We observed that the expression of miR-29a and miR-29b increased significantly in patients with ONS-PR at 3 months in comparison with baseline, stabilizing at the sixth month. 

Regarding the expression levels of *TGFB1*, we observed a decrease in this transcription factor in both ONS groups at 3 months with respect to the control. Consistent with these results, several studies have emphasized the role of *TGFB1* in the induction of renal fibrosis and the progression of CKD by promoting inflammation and ECM modulation [32,33]. Moreover, *RUNX2*, well known for its crucial role in osteoblast differentiation and bone formation, has also been shown to be involved in fibrosis induction in various organs [34,35]. Furthermore, some studies have reported a tight correlation between the *RUNX* family and *TGFB1* with fibrosis and CKD development [36,37,38,39]. In our study, we observed a decrease in *RUNX2* expression in the ONS-PR group compared to the control group at 3 months, followed by a slight decrease at 3 months compared to baseline and a subsequent increase at the 6-month intervention. In addition, Wang and collaborators have revealed that miR-29a attenuates kidney fibrosis in Unilateral Ureteral Obstruction (UUO) mice by inhibiting fibrotic proteins and *TGFB1* [17]. Moreover, in vitro upregulation of miR-29b inhibited the expression of *TGFB1* [30]. These observations are in accordance with our results, suggesting that these microRNAs could exert protective effects in the context of CKD by inhibiting a pro-fibrotic state mediated by *TGFB1* and *RUNX2.*


Concerning *PTEN* expression, we found that it was significantly elevated in the ONS-PR group at 3 months in comparison with the control group. Higgins and collaborators have demonstrated that *PTEN* prevents kidney fibrosis and renal injury by inhibiting PI3K/Akt and by counteracting the fibrotic effect of *TGFB1* in cultured renal cells and the UUO model [40]. In addition, another study has revealed that increased *TGFB* expression during acute kidney injury (AKI) was associated with the loss of *PTEN* expression [41]. Also, the recovery of *PTEN* was accompanied with normal tubule repair and less fibrosis. Furthermore, inhibition of *PTEN* expression increased renal fibrosis in an AKI mice model [36]. Additionally, other researchers have found that *PTEN* improves cellular fibrotic changes and renal fibrosis via inhibiting the FAK/AKT signaling pathway [42]. In line with these results, one study has shown that Homeobox transcript antisense RNA (HOTAIR) downregulates *PTEN* via miR-29b inhibition during liver fibrosis [43]. Moreover, another study has revealed that miR-29b contributed to the regulation of renal interstitial fibrosis via targeting the PI3K/Akt signaling pathway and *PTEN* expression [44]. These findings suggest the role that miR-29a and b and its target genes could play in kidney fibrosis and renal injury within CKD. 

In relation to *LEPTIN* levels, the increase in its expression in the ONS-PL group raises interesting questions about the possible link between the oral supplement and appetite regulation in malnourished patients. Leptin is a peptide hormone primarily synthesized in adipose tissue and plays an important role in regulating appetite and energy balance [45]. Moreover, leptin exerts a pivotal role in inducing angiogenesis and adipogenesis in adipose tissue [46]. In addition, it has been described that low levels of leptin are associated with protein energy wasting in people with CKD, hemodialysis and with a worse prognosis [47,48,49], even though a correlation with the amount of fat mass is maintained [50]. In our previous study [2], both groups that received the ONS had increased weight at the end of the intervention, improving the nutritional status at the expense of especially fat-free mass but also with an increase in fat mass (around 1 kg). Furthermore, dietary total fat increased significantly especially in the ONS-PL group [2]. In this context, the significant increase in fat intake could have stimulated the *LEPTIN* expression as an appetite-regulating response also associated with the increase in body fat mass. Furthermore, the probiotics in the ONS-PR group may have influenced the gut microbiota composition, mitigating a significant leptin response through various mechanisms [51]. 

The study’s primary limitation stemmed from a relatively high dropout rate, largely attributed to challenges arising from the SARS-CoV-2 pandemic and the associated difficulties in ensuring continued patient follow-up [2], which has contributed to not finding significant changes in other circulating miRNAs. Caution is warranted in interpreting the results of this study, as the limited sample size of patients examined may impact the generalizability and robustness of the findings. As strengths of the study, we highlight the fact that it is a randomized clinical trial (double-blind regarding probiotics intake), the follow-up is long-term (6 months), it has a multicentric nature, the measurement of multiple parameters was conducted (including biomarkers, miRNAs and their target genes) and there was a comparison with a control group on individualized diets prescribed by registered dietitians.

## 4. Materials and Methods

### 4.1. Design

The trial design is set out in the previous publication [2]. Briefly, it is a randomized, multicenter, parallel-group trial with 3 groups, open regarding the intake of ONS or individualized diet recommendations but double-blind for the intake of probiotics. Patients were randomized to one of the following three groups (using a computer-generated random number table): 

1: Control (C): received individualized dietary recommendations (IDRs). 

2: ONS + placebo (ONS-PL): received ONS + IDR.

3: ONS + probiotics (ONS-PR): received ONS with probiotics + IDR.

The Renacare^®^ ONS was developed for malnourished hemodialysis patients. It is a high-calorie (2 kcal/mL) and high-protein supplement enriched with functional nutrients (extra virgin olive oil, omega 3 fatty acids, whey protein, antioxidants, low-glycemic index carbohydrates, fiber and carnitine). The composition is shown on the website “https://adventiapharma.com/nutricion-clinica/productos/enteral-oral/bi1-renacare-dial (accessed on 15 October 2023)”. 

Inclusion and exclusion criteria are detailed in our previous publication [2]. Participants included adult subjects (>18 years) undergoing hemodialysis (standard therapy or on-line hemodiafiltration with high reinfusion rate therapy not being modified in the 3 months before inclusion for more than 6 months before inclusion) and who were malnourished. Written informed consent was obtained from all patients.

The nutritional requirements were assessed according to the recommendations of the International Society of Renal Nutrition and Metabolism. Target protein intake was at least 1.2 g/kg/day [52]. All participants had a face-to-face interview with a nutritionist at baseline and after 3 and 6 months. Patients randomized to the ONS group were advised to consume 2 bricks of 400 mL per day (minimum 1 day—200 mL). The daily intake of ONS was prospectively recorded by patients on a data collection sheet. Probiotics and the placebo were supplied in visually indistinguishable capsule form (one capsule 380g). Each probiotic capsule contains live bacteria: *Bifidobacterium breve* CNCM I-4035 (1.00 × 10^9^ Colony Forming Units, CFUs), *Bifidobacterium animalis lactis* BPL1 CECT 8145 (3.50 × 10^9^ CFU) and *Lactobacillus paracasei* CNCM I-4034 (5.00 × 10^8^ CFU). 

The Research Ethics Committee of the Province of Malaga approved the study and the protocol complies with the ethical standards of the Declaration of Helsinki. The study was registered under the code NCT03924089.

### 4.2. Outcomes

Examinations were performed at baseline and after 3 and 6 months. Fasting blood samples were drawn before beginning the dialysis session; plasma and serum were separated into aliquots and stored until analysis at −80 °C in the Hospital-IBIMA biobank that also belongs to the Andalusian Public Health System Biobank, belonging to the National Biobank Platform (exp. PT20/00101).

### 4.3. miRNA and RNA Extraction

miRNAs were extracted from plasma samples by means of automated method with a Maxwell^®^ 16 miRNA Tissue Kit (Promega Biotech Ibérica S.L., Madrid, Spain) according to the manufacturer’s protocol. For each sample, 300 μL of plasma was consequently mixed with the following reagents: 200 μL of homogenization solution, 200 μL of lysis buffer and 15 μL of Proteinase K. Samples were incubated for 20 min on a heat block at 37 °C, 300 rpm. After the incubation, samples were transferred to RSC cartridges followed by automated RNA extraction with the Maxwell instrument. Finally, total RNA was eluted in 30 μL nuclease-free water and stored at −80 °C until use.

Total RNA was extracted from PAXgene Blood RNA Tubes (Qiagen, Hilden, Germany). Briefly, the content of the thawed tube was transferred to a 50 mL conical tube with 3 mL 1× phosphate buffered serum (PBS), then vortexed for 30 s and later centrifuged at 4000× *g* at 4 °C for 20 min. The RNA pellet was further lysed with QIAzol Lysis reagent (Qiagen, Hilden, Germany ), followed by incubation with 200 μL chloroform. After the phase separation, the upper aqueous phase was mixed with isopropanol and incubated for 10 min on ice. After centrifugation, the RNA pellet was washed with 75% ethanol. After centrifugation, the RNA pellet was air-dried at room temperature for 10 min. Total RNA was resuspended in 25 μL of nuclease-free water and stored at −80 °C until use. 

The purity of miRNA and total RNA was determined by the absorbance 260/280 ratio on a Nanodrop ND-2000 spectrophotometer (Thermo Fisher Scientific Inc., Waltham, MA, USA). 

### 4.4. RT-qPCR

cDNA was synthesized from miRNAs and mRNAs by reverse transcription with the Universal cDNA Synthesis Kit (Exiqon, Vedbaek, Denmark) following the manufacturer’s recommendations for each kit. 

Measurements of miRNA expression levels (miR-21-5p, miR-29a-3p, miR-29b-3p, miR-126-3p, miR-128-3p, miR-155-5p, miR-233-3p and miR-378-3p) were performed in duplicate by real-time qPCR in 384 plates in a LightCycler 480 (Roche Diagnostics, S.L, Barcelona, Spain) at the Genomics platform of IBIMA-Plataforma Bionand. The master mix was prepared following the guidelines of Exiqon with GoTaq^®^ qPCR Master Mix (Promega Biotech Ibérica S.L., Madrid, Spain) and the specific LNA™ miRNA PCR primer Assy (Exiqon, Vedbaek, Denmark) (Table 3).

Gene expression levels (TBP, *TGFB*, *RUNX2*, *PTEN*, *AKT1*, *RELA*, *TNF*, *IL1B*, *IL6*, *MYH7B*, *HEF2C*, *CTNNB1*, *LEP*, *GHRL*, *RASA1* and *PPARG*) were analyzed in duplicate by qPCR using a QuantStudio 12K Flex Real Time PCR 384 system (Thermo Fisher Scientific Inc., Waltham, MA, USA) with TaqMan technology. Predesigned Thermo Fisher primers were used to detect the canonical isoform of the gene of interest in humans, built based on the GRCh38 reference genome (Table 4). These primers have been validated in silico and in vitro by Thermo Fisher (https://www.thermofisher.com/taqman/gene-expression/assay, accessed on 10 November 2022). To carry out the qPCR assays, the recommendations established by the manufacturer were followed, using the TaqMan^®^ Fast Advanced Master Mix. 

Quantitative measures were obtained using the 2^−ΔCt^ method. Expression results were given as the expression ratio relative to miR-93 for miRNAs after an analysis with RefFinder (https://blooge.cn/RefFinder/, accessed on 17 February 2023), as it was found to be the most stable miRNA. Furthermore, Donati et al. underscored the stability and suitability of this microRNA as an endogenous control in plasma [53]. TBP was used as the normalizing gene for gene expression, selected after performing a screening (TaqMan^TM^ Array Human Endogenous Controls Plate, Fast 96-well, Thermo Fisher Scientific Inc., Waltham, MA, USA) with 32 possible constituent genes in triplicate, in a study subpopulation made up of 3 subjects from each of the 3 groups, a total of 9 subjects. Also, in order to detect the most stable gene, the RefFinder program was used.

For data normalization, negative controls and a specific calibrator in all plates for interplate normalization were included. Those determinations with Ct > 38 were considered under the limit of detection and were taken out of the analysis.

### 4.5. In Silico Identification of Predicted and Validated miRNA Target Genes

The miRTarBase 9.0 (https://mirtarbase.cuhk.edu.cn/~miRTarBase/miRTarBase_2022/php/index.php, accessed on 30 March 2023), TarBase v.8 (https://dianalab.e-ce.uth.gr/html/diana/web/index.php?r=tarbasev8, accessed on 30 March 2023) and TargetScan 8.0 (https://www.targetscan.org/vert_80/, accessed on 30 March 2023) databases were used to identify the validated and predicted target genes (TGs) of miR-29a and miR-29b. The Protein Analysis Through Evolutionary Relationships (PANTHER) Classification System (http://www.pantherdb.org/, accessed on 30 March 2023) and the Database for Annotation, Visualization and Integrated Discovery (DAVID; https://david.ncifcrf.gov/, accessed on 30 March 2023) were used to annotate the biological processes and pathways of the predicted TGs. Then, we focused our study on TGs involved in inflammation, lipid metabolism and bone formation, as these are associated with our ONS and renal function. The interactions between miRNAs and the TGs were visualized with Cytoscape software v.3.10.1 (https://cytoscape.org/, accessed on 30 November 2023).

### 4.6. Biomarkers

Fasting blood samples were drawn before beginning the dialysis session and plasma and serum were separated into aliquots and stored until analysis at −80 °C in the Hospital-IBIMA biobank. IL-10, IL-8 and TNFα were measured in 25 μL of serum with ProcartaPlex multiplex Immunoassay (Thermo Fisher Scientific, Waltham, MA, USA) following manufacturer’s instructions [2]. All measurements were performed in duplicate and serum concentrations were obtained using a standard curve.

### 4.7. Statistical Analysis

Data analysis was carried out using the IBM SPSS Statistics version 26 (IBM Corp. Released 2019, IBM SPSS Statistics for Windows, Version 26.0, Armonk, NY, USA) program. Data are expressed as the mean ± SEM. Comparisons among multiple groups were performed by one-way ANOVA, and comparisons between two groups were performed by Student’s *t* test. To compare variables according to the group of study and the modifications along time (baseline, 3 and 6 months), ANOVA for repeated variables was used. The level of significance taken into account was 5%. For multiple comparisons (post hoc), Bonferroni’s correction was considered. The percentage of the increased or decreased expression of miRNAs or serum parameters with respect to the basal levels were calculated as follows:

Percentage increase/decrease = ((log parameter at 3 months-log parameter at basal time)/log parameter at basal time) × 100

Correlations were analyzed using Pearson’s correlation coefficient using the percentage change between times of intervention between the groups. Normality was assessed by the Shapiro–Wilk test and all the variables that did not present normal distribution (*p* > 0.05) were log transformed for normalization; thus, we performed parametric tests for our analysis. The miRNAs’ expression graphics were created using GraphPad Prism version 9.0.0 for Windows, GraphPad Software (Boston, MA, USA). 

## 5. Conclusions

The new oral nutritional supplement specifically designed for malnourished (or at risk) hemodialysis patients with a “similar to the Mediterranean diet” pattern, associated with probiotics, increases the expression levels of miR-29a and miR-29b after 3 months of intervention, modifying the expression of target genes with anti-inflammatory and anti-fibrotic actions. 

This study highlights the potential benefit of this oral nutritional supplement, especially associated with probiotics, in malnourished patients with chronic renal disease on hemodialysis.

## Figures and Tables

**Figure 1 ijms-25-01132-f001:**
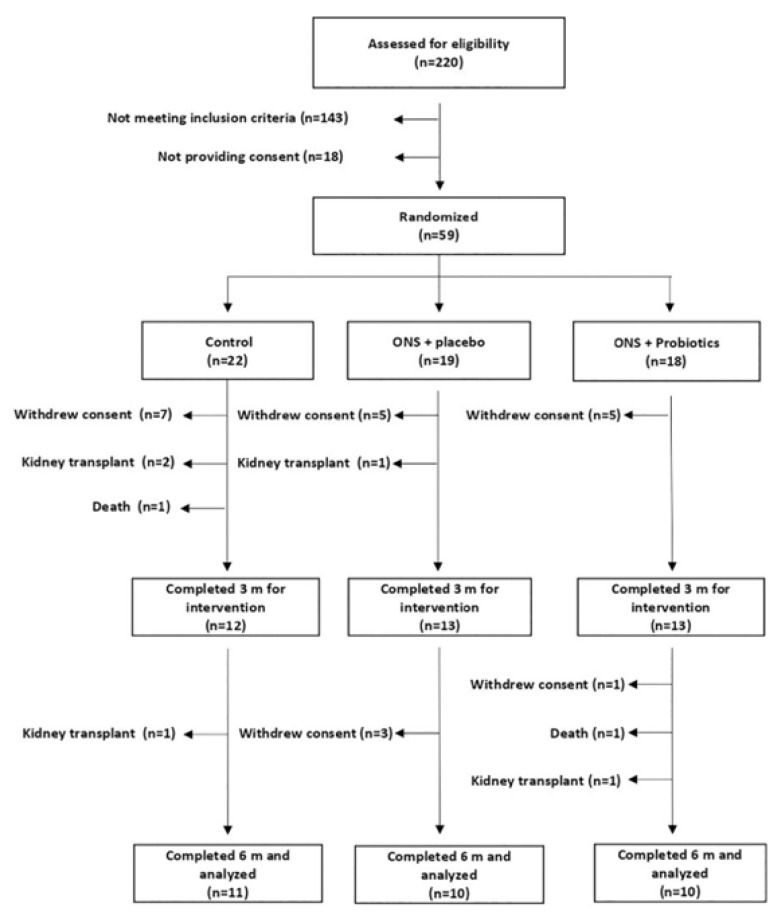
Study flowchart. ONS: Oral nutritional supplement.

**Figure 2 ijms-25-01132-f002:**
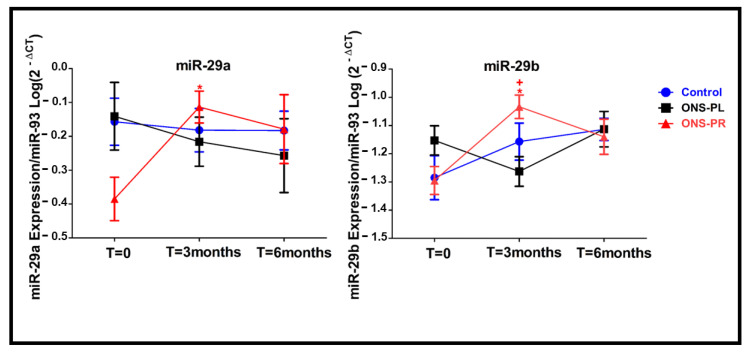
*MiR-29a and miR-29b expression levels.* Examinations were carried out at baseline and after 3 and 6 months. All the values come from the logarithm of the correspondent parameter. miR-93 was used as the reference gene. Data are presented as mean ± standard deviation. * *p* < 0.05 versus baseline. + *p* < 0.05 versus control. ONS-PL = oral nutritional supplement with placebo; ONS-PR = oral nutritional supplement with probiotics. Note: the negative values in miRNAs’ expression result from the fact that the 2^−ΔCt^ value is lower than 1.

**Figure 3 ijms-25-01132-f003:**
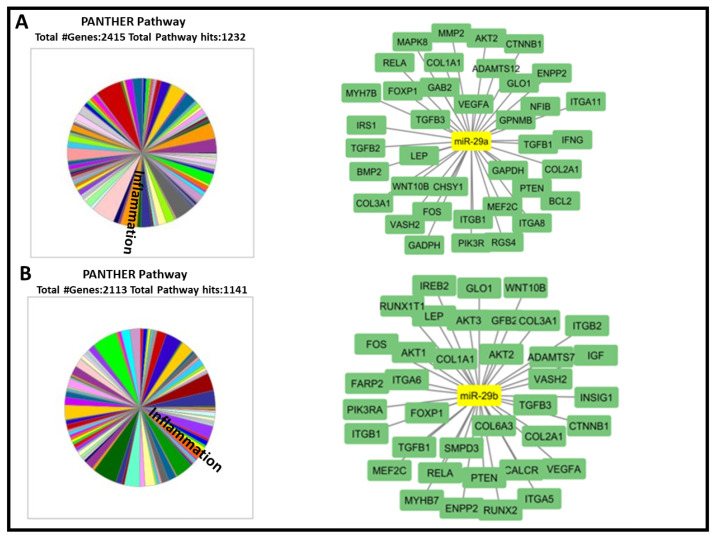
In silico analysis and schema recapitulating the interactions between miR-29a, miR-29b and TGs involved in inflammation, lipid metabolism and bone formation. (**A**,**B**) miRTarbase 9.0, Tarbase v.8 and targetScan 8.0 were used to identify possible TGs of miR-29a and miR-29b (yellow nodes). Panther and DAVID databases were used to identify TGs (green nodes) implicated in inflammation, lipid metabolism and bone formation. The interactions among miRNAs and TGs were visualized using Cytoscape v.3.10.1.

**Figure 4 ijms-25-01132-f004:**
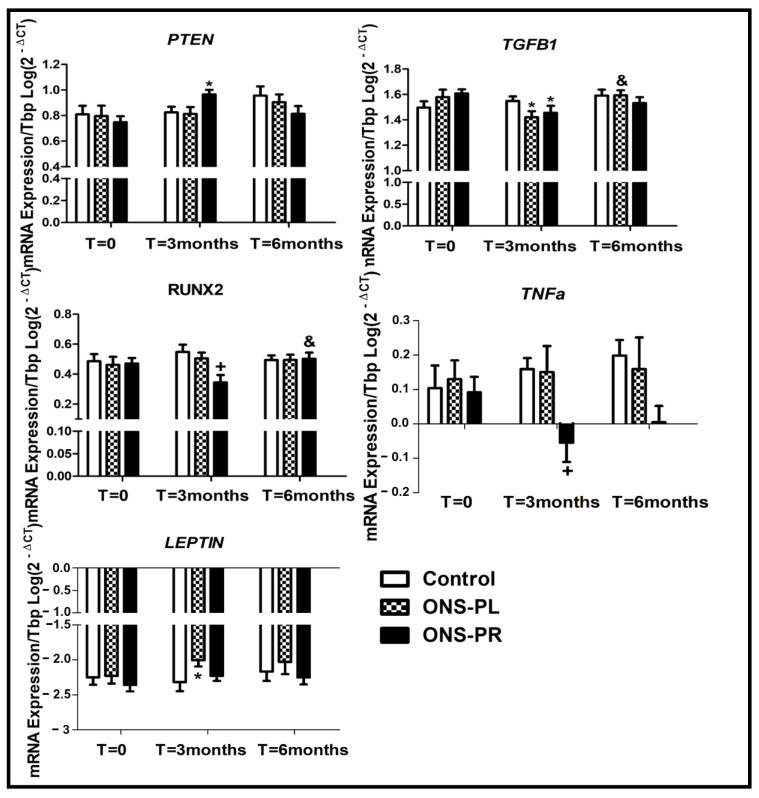
*mRNA expression levels from human blood*. Examinations were carried out at baseline and after 3 and 6 months. All the values come from the logarithm of the correspondent parameter. * *p* < 0.05 versus baseline. + *p* < 0.05 versus control. & *p* < 0.05 versus 3-month intervention. ONS-PL = oral nutritional supplement with placebo; ONS-PR = oral nutritional supplement with probiotics. *PTEN*: Phosphatase and tensin homolog, *TGFB1:* Transforming growth factor beta 1, *RUNX2*: Runt-related transcription factor 2, *LEPTIN*, *TNFa*: Tumor necrosis factor a.

**Figure 5 ijms-25-01132-f005:**
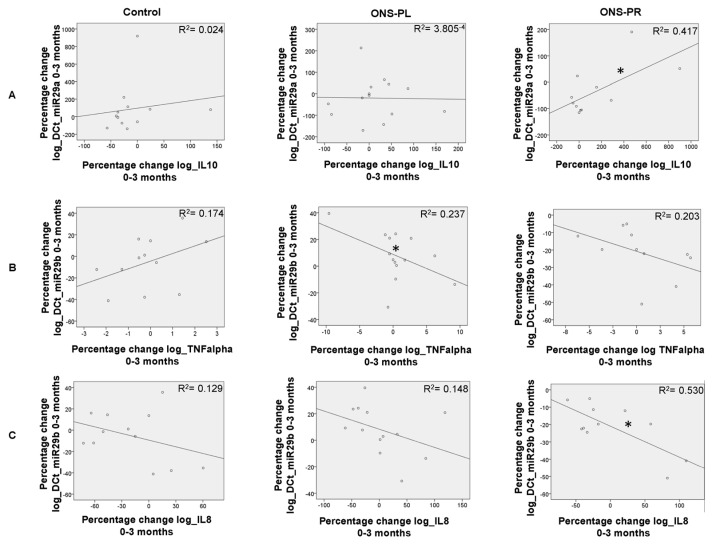
*Correlations between miR-29a and miR-29b expression levels and serum biomarkers*. Examinations were carried out at baseline and after 3 months. Correlations were analyzed using Pearson’s correlation coefficient using the percentage change of the logarithm of the variables between 3 months and baseline in each group. (**A**) Correlation between miR-29a and IL-10. (**B**) Correlation between miR-29b and TNFalpha. (**C**) Correlation between miR-29b and IL-8. ONS-PL = oral nutritional supplement with placebo; ONS-PR = oral nutritional supplement with probiotics. * *p* < 0.05.

**Table 1 ijms-25-01132-t001:** Baseline patient characteristics.

	Control	ONS-PL	ONS-PR	*p*-Values
	(*n* = 11)	(*n* = 10)	(*n* = 10)	
Age in years m ± ds	76.3 ± 8.7	65.1 ± 18.4	66 ± 18.5	ns
Sex, men % (*n*)	73 (8)	80 (8)	70 (7)	ns
Body mass index (kg/m^2^)	25.1 ± 3	23.4 ± 5	25.2 ± 5	ns
Diabetes mellitus % (*n*)	36.4 (4)	40 (4)	30 (3)	ns
Antibiotic treatment in the past month	9.1 (1)	20 (2)	10 (1)	ns
Yoghurt or fermented milk consumption in the last month % (*n*)	63.6 (7)	60 (6)	60 (6)	ns
Charlson Comorbidity Index. m ± ds	4 ± 2.31	5.1 ± 2.02	4.18 ± 2.6	ns

Data are presented as mean ± standard deviation (m ± ds) or in percentage % (*n*), ONS-PL supplement group + placebo; ONS-PR supplement group + probiotics. ns: not statistically significant.

**Table 2 ijms-25-01132-t002:** Panther and DAVID bioinformatic analysis. Biological processes associated with the TGs of miR-29a and miR-29b.

MiRNA	Database	Biological Process	Genes
MiR-29a	Panther	Inflammation	IFNG, ITGB1, **RELA**, RGS4
	DAVID	Osteoclast differentiation	**TGFB1**, GLO1, PIK3R1, FOS, GAB2, FOXP1, BMP2, CALCR, **CTNNB1**
		Chondrocyte differentiation	WNT10B, **MEF2C**, **TGFB1**, COL3A1, BMP2, COL2A1, NFIB, **CTNNB1**, ADAMTS12
		Ossification	**MEF2C**, SFRP1, **TGFB1**, CALCR, CHSY1, BCL2
		Cardiomyopathy	IRS1, **PTEN**, **RELA**, MAPK8, AKT2, TGFB2, TGFB1, TGFB3, MMP2, **MYH7B**, COL1A1, COL3A1, GAPDH
		Regulation of angiogenesis	GPNMB, **LEP**, ENPP2, VASH2, **CTNNB1**, VEGFA
		Cell–matrix adhesion	ADAMTS12, COL3A1, COL2A1, ITGA11, ITGA8, ADAM9, **CTNNB1**
MiR-29b	Panther	Inflammation	ITGB1, **RELA**, **AKT1**, AKT2, AKT3, COL6A3
	DAVID	Osteoclast differentiation	FARP2, **TGFB1**, CALCR, GLO1, IREB2, **CTNNB1**, FOS, PIK3R1, FOXP1
		Chondrocyte differentiation	WNT10B, **MEF2C**, COL3A1, COL2A1, **TGFB1**, **RUNX2**, ADAMTS7
		Ossification	SMPD3, COL1A1, **MEF2C**, COL2A1, **RUNX2**
		Cardiomyopathy	ITGB1, TGFB2, **TGFB1**, **MYH7B**, TGFB3, IGF, **RELA**, AKT2, AKT3, ITGA6, ITGA5, **AKT1**
		Regulation of angiogenesis	**LEP**, ENPP2, VASH2, **CTNNB1**, **PTEN**, VEGFA
		Cell–matrix adhesion	FGB, ITGB1, COL3A1, COL5A3, ITGA11, **CTNNB1**, ITGA6, ITGA5
		Regulation of fat cell differentiation	WNT10B, **TGFB1**, INSIG1, RUNX1T1

Nine important target genes that regulate two or more of processes were selected for analysis (highlighted in bold).

**Table 3 ijms-25-01132-t003:** Primers used for amplification miRNAs: miRNA quantification using LNA-optimized, SYBR^®^ Green-based miRNA PCR.

miRNA Assay ID	miRCURY LNA miRNA PCR Assays	Mature miRNA Sequence	miRBase Accession
hsa-miR-21-5p	YP00204230	5′UAGCUUAUCAGACUGAUGUUGA	MIMAT0000076 (https://www.mirbase.org/mature/MIMAT0000076, accessed on 15 October 2023)
hsa-miR-29a-3p	YP00204698	5′UAGCACCAUCUGAAAUCGGUUA	MIMAT0000086 (https://www.mirbase.org/mature/MIMAT0000086, accessed on 15 October 2023)
hsa-miR-29b-3p	YP00204679	5′UAGCACCAUUUGAAAUCAGUGUU	MIMAT0000100 (https://www.mirbase.org/mature/MIMAT0000100, accessed on 15 October 2023)
hsa-miR-93-5p	YP00204715	5′CAAAGUGCUGUUCGUGCAGGUAG	MIMAT0000093 (https://mirbase.org/mature/MIMAT0000093, accessed on 15 October 2023)
hsa-miR-126-3p	YP00204227	5′UCGUACCGUGAGUAAUAAUGCG	MIMAT0000445 (https://mirbase.org/mature/MIMAT0000445, accessed on 15 October 2023)
hsa-miR-128-3p	YP00205995	5′UCACAGUGAACCGGUCUCUUU	MIMAT0000424 (https://mirbase.org/mature/MIMAT0000424, accessed on 15 October 2023)
hsa-miR-155-5p	YP02119311	5′UUAAUGCUAAUCGUGAUAGGGGUU	MIMAT0000646 (https://mirbase.org/mature/MIMAT0000646, accessed on 15 October 2023)
hsa-miR-223-3p	YP00205986	5′UGUCAGUUUGUCAAAUACCCCA	MIMAT0000280 (https://mirbase.org/mature/MIMAT0000280, accessed on 15 October 2023)
hsa-miR-378a-3p	YP00205946	5′ACUGGACUUGGAGUCAGAAGGC	MIMAT0000732 (https://mirbase.org/mature/MIMAT0000732, accessed on 15 October 2023)

**Table 4 ijms-25-01132-t004:** Primers used for amplification: genes.

Gene	Ref_Seq	Assay_ID	Dye Label	Chromosome Location
TBP	NM_001172085.1 (http://www.ncbi.nlm.nih.gov/nuccore/NM_001172085.1, accessed on 15 October 2023)	Hs00427620_m1	FAM-MGB	Chr.6 170554333–170572870
TGFB1	NM_000660.5 (http://www.ncbi.nlm.nih.gov/nuccore/NM_000660.5, accessed on 15 October 2023)	Hs00998133_m1	FAM-MGB	Chr.19: 41330531-41353933
RUNX2	NM_001015051.3 (http://www.ncbi.nlm.nih.gov/nuccore/NM_001015051.3, accessed on 15 October 2023)	Hs01047973_m1	FAM-MGB	Chr.6: 45328142-45664032
PTEN	NM_000314.6 (http://www.ncbi.nlm.nih.gov/nuccore/NM_000314.6, accessed on 15 October 2023)	Hs02621230_s1	FAM-MGB	Chr.10: 87863438-87971930
AKT1	NM_001014431.1 (http://www.ncbi.nlm.nih.gov/nuccore/NM_001014431.1, accessed on 15 October 2023)	Hs00178289_m1	FAM-MGB	Chr.14: 104769349-104795743
RELA	NM_001145138.1 (http://www.ncbi.nlm.nih.gov/nuccore/NM_001145138.1, accessed on 15 October 2023)	Hs01042014_m1	FAM-MGB	Chr.11: 65653596-65662972
TNF	NM_000594.3 (http://www.ncbi.nlm.nih.gov/nuccore/NM_000594.3, accessed on 15 October 2023)	Hs00174128_m1	FAM-MGB	Chr.6: 31575567-31578336
IL1 B	NM_000576.2 (http://www.ncbi.nlm.nih.gov/nuccore/NM_000576.2, accessed on 15 October 2023)	Hs01555410_m1	FAM-MGB	Chr.2: 112829758-112836842
IL6	NM_000600.4 (http://www.ncbi.nlm.nih.gov/nuccore/NM_000600.4, accessed on 15 October 2023)	Hs00174131_m1	FAM-MGB	Chr.7: 22725889-22732002
MYH7B	NM_020884.4 (http://www.ncbi.nlm.nih.gov/nuccore/NM_020884.4, accessed on 15 October 2023)	Hs00293096_m1	FAM-MGB	Chr.20: 34955835-35002437
MEF2C	NM_001131005.2 (http://www.ncbi.nlm.nih.gov/nuccore/NM_001131005.2, accessed on 15 October 2023)	Hs00231149_m1	FAM-MGB	Chr.5: 88718241-88904105
CTNNB1	NM_001098209.1 (http://www.ncbi.nlm.nih.gov/nuccore/NM_001098209.1, accessed on 15 October 2023)	Hs00355045_m1	FAM-MGB	Chr.3: 41199451-41240448
LEP	NM_000230.2 (http://www.ncbi.nlm.nih.gov/nuccore/NM_000230.2, accessed on 15 October 2023)	Hs00174877_m1	FAM-MGB	Chr.7: 128241201-128257629
GHRL	NM_001134941.2 (http://www.ncbi.nlm.nih.gov/nuccore/NM_001134941.2, accessed on 15 October 2023)	Hs01074053_m1	FAM-MGB	Chr.3: 10285750-10292947
RASA1	NM_002890.2 (www.ncbi.nlm.nih.gov/nuccore/NM_002890.2, accessed on 15 October 2023)	Hs00243115_m1	FAM-MGB	Chr.5: 87267801-87391926
PPARG	NM_005037.5 (http://www.ncbi.nlm.nih.gov/nuccore/NM_005037.5, accessed on 15 October 2023)	Hs01115513_m1	FAM-MGB	Chr.3: 12287850-12471054

## Data Availability

The data that support the findings of this study are available from the corresponding author upon reasonable request.

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
