# Peer review of "Modulation of miR-29a and miR-29b Expression and Their Target Genes Related to Inflammation and Renal Fibrosis by an Oral Nutritional Supplement with Probiotics in Malnourished Hemodialysis Patients"

_ijms, 2024, doi:10.3390/ijms25021132_

Round 1

Reviewer 1 Report

Comments and Suggestions for Authors

Dear Editor

In the present study, a modified diet similar to the Mediterranean diet with probiotics was used in malnourished hemodialysis patients. In the first stage, the authors examine the changes of several miRs as blood biomarkers. They notice stable changes in miR 29a and b after 3 months. By examining the target genes of miR 29a & b, they found that this type of diet alone (in some cases) or together with probiotics was able to improve the expression of inflammatory genes and genes involved in renal fibrosis. Therefore, they suggest this diet + probiotics in patients with chronic kidney disease and undergoing hemodialysis.

Considering the choice of the type of disease, this study is important. Comments are provided here:

-It is suggested to report changes in genes related to inflammation and renal fibrosis in the title of the manuscript.

- The number of patients in each group should be stated in the abstract

-Because the number of samples in each group is small. If the variables do not show a significant difference between the groups at the baseline level, it seems that the comparison with the baseline in each group should be reported first, and then with the control group.

- Why is the demographic table of patients not reported? Age, gender, weight of patients, how long they have been undergoing dialysis and other variables...

- Although the examination of blood biomarkers in this study is very valuable, did the patients show a better condition in kidney function parameters by receiving this combined diet? Because a patient undergoing hemodialysis, the first issue is to help improve the condition of body fluids.

- It should be determined for which biomarker, serum and for which biomarkers, plasma was used? Why are both used?

- In Figure 2, the role of miR 93 in the relative report of miR 29 a & b should be specified. Drawing the graph in reverse to show negative values is confusing.

- According to the caption of Figure 4, the target genes involved in inflammation, fat metabolism and bone formation have been identified. Have been the target genes involved in fat metabolism selected in this study?

-In the examination of the expression of all genes in the control group, an increasing trend is observed up to 6 months. Is it not important that the usual diet is not suitable for hemodialysis patients?

Comments on the Quality of English Language

Minor editing of English language required

Author Response

We appreciate the reviewer´s comments on our study. Please find the detailed responses attached and the corresponding revisions/corrections highlighted in red in the re-submitted files. Thank you very much for taking the time to review this manuscript and for giving us the opportunity to improve it. We remain at your disposal for any further comments.

Reviewer 2 Report

Comments and Suggestions for Authors

The article is interesting for the journal and should be published, but requires a broader description of the examined patients and correction of several other minor comments

1. the title is too long, try shortening it

2. add the license in the left margin of the first page

3. the number of patients is small, which is a serious limitation of the study, therefore the results should be treated as pilot studies. on the other hand, the survival of malnourished CKD patients may preclude long-term follow-up.

4. the materials and methods lack a description of the studied group of patients, anthropometric and biochemical parameters that should be presented in comparison also between the three (control, ONS-placebo, ONS-probiotic).

5. These parameters should be presented both before and after the intervention

6. Good nutritional status is described as having a BMI of 23–26. Protein and energy malnutrition is present and anorexia occurs, depending on the source, in up to 50% of CKD stage 5 patients.

https://doi.org/10.3390/nu15040860

What was the problem like in the analyzed group?

7. English requires minor correction

8. conclusions should be cautious due to the small number of patients examined

Comments on the Quality of English Language

7. English requires minor correction

Author Response

(The authors gave the same response as above.)

Reviewer 3 Report

Comments and Suggestions for Authors

The manuscript titled: “Modulation of miR-29a and miR-29b expression by an oral nutritional supplement similar to the Mediterranean diet pattern with probiotics in malnourished hemodialysis patients” (Manuscript ID ijms-2797130) describes the role of a new oral nutritional supplement (ONS) in malnourished hemodialysis patients. The authors found that the new Oral Nutritional Supplement associated with probiotics increases the expression levels of miR-29a and miR-29b after three months of intervention, modifying the expression of target genes with anti-inflammatory and anti-fibrotic actions.

1.       The study groups are relatively small.

2.       Recent data describe the influence of various nutritional patterns on the diet. Which of the components of oral nutritional supplement (ONS) - as extra virgin olive oil, n-3 faĴy acids, whey protein, fiber, and antioxidants have the highest influence on the expression levels of miR-29a and miR-29b and leptin expression?

3.       What is the exact content of specific nutrients in ONS?

4.       Did the mechanism of expression of these genes in malnourished hemodialysis patients differ from the general population? Why?

5.       Can ONS supplement be used in another group of malnourished patients – e.g., oncologic patients or patients with mucoviscidosis?

6.       Do the amount and type of probiotics influence gene expression in your study?

Author Response

(The authors gave the same response as above.)

Round 2

Reviewer 1 Report

Comments and Suggestions for Authors

To Authors

The authors have convincingly answered the questions and ambiguities. Regarding comment 5, the data and results should be added to the manuscript.

I have no further comment. It is appropriate to publish.

Comments on the Quality of English Language

I have no further comment

Author Response

Thank you very much for taking the time to read our responses. We greatly appreciate the decision to accept the manuscript for publication. According to your suggestion on comment 5, we have added the following on page 3, lines 123-127: “We also observed a slight decrease in extracellular water measured by bioimpedance, but without significant differences between groups or to the baseline value (data not shown). Moreover, all these body composition changes occurred without an increase in serum phosphorus or potassium levels (with a clear tendency to decrease in the supplemented groups) [2]”.

Reviewer 2 Report

Comments and Suggestions for Authors

The Authors responded to the comments and provided an explanation

thank you for your cooperation

Author Response

Thank you very much for taking the time to read our responses.